# Salt Induces Adipogenesis/Lipogenesis and Inflammatory Adipocytokines Secretion in Adipocytes

**DOI:** 10.3390/ijms20010160

**Published:** 2019-01-04

**Authors:** Myoungsook Lee, Sungbin Richard Sorn, Yunkyoung Lee, Inhae Kang

**Affiliations:** 1Department of Food and Nutrition, Sungshin Women’s University, Seoul 01133, Korea; 2Research Institute of Obesity Sciences, Sungshin Women’s University, Seoul 01133, Korea; 3College of Medicine, The Catholic University of Korea, Seoul 06591, Korea; rssorn7@catholic.ac.kr; 4Department of Food Science and Nutrition, Jeju National University, Jeju 63243, Korea; lyk1230@jejunu.ac.kr (Y.L.); inhaek@jejunu.ac.kr (I.K.)

**Keywords:** salt, obesity, adipogenesis, inflammatory cytokines, MAPK/ERK, Akt-mTOR

## Abstract

It is well known that high salt intake is associated with cardiovascular diseases including hypertension. However, the research on the mechanism of obesity due to high salt intake is rare. To evaluate the roles of salt on obesity prevalence, the gene expression of adipogenesis/lipogenesis and adipocytokines secretion according to adipocyte dysfunction were investigated in salt-loading adipocytes. High salt dose-dependently increased the expression of adipogenic/lipogenic genes, such as *PPAR-γ, C/EBPα*, *SREBP1c, ACC, FAS,* and *aP2*, but decreased the gene of lipolysis like *AMPK*, ultimately resulting in fat accumulation. With SIK-2 and Na^+^/K^+^-ATPase activation, salt increased the metabolites involved in the renin-angiotensin-aldosterone system (RAAS) such as *ADD1, CYP11β2,* and *MCR.* Increasing insulin dependent insulin receptor substrate (IRS)-signaling, resulting in the insulin resistance, mitogen-activated protein kinase/extracellular signal-regulated kinase (MAPK/ERK) and Akt-mTOR were activated but AMPK(Thr^172^) was depressed in salt-loading adipocytes. The expression of pro-inflammatory adipocytokines, TNFα, MCP-1, COX-2, IL-17A, IL-6, leptin, and leptin to adiponectin ratio (LAR) were dose-dependently increased by salt treatment. Using the inhibitors of MAPK/ERK, U0126, we found that the crosstalk among the signaling pathways of MAPK/ERK, Akt-mTOR, and the inflammatory adipogenesis can be the possible mechanism of salt-linked obesity. The possibilities of whether the defense mechanisms against high dose of intracellular salts provoke signaling for adipocytes differentiation or interact with surrounding tissues through other pathways will be explored in future research.

## 1. Introduction

Metabolic dysfunction of adipocytes associated with obesity-linked complications on adipogenesis/lipogenesis and adipocytokines production positively depended on various obesogenic environments such as excessive consumption of fat, salt and alcohol, sedentary lifestyles, stresses, and so on [1,2]. According to the 2010 Korea National Health and Nutrition Examination Survey (KNHANES), the average daily sodium intake (5000 mg/day; salt 13.8 g/day) of Korean adults is 2.5-fold higher than that recommended by the World Health Organization (WHO) [3]. In the last 10 years, an increase in salt intake has been shown to be closely associated with an obesity prevalence in Korean children and adults. Salt intake leads to an increase in blood volume and dehydration of cells, and the complex hormonal association eventually drives body to regulate the extracellular volume [4]. However, the salt-induced mechanisms or pathways related to adipogenesis/lipogenesis and adipocytokines secretion, and the appropriate level of salt intake for the homeostasis of water and sodium, are not currently well known.

A well-known mechanism, the alteration in renin-angiotensin-aldosterone system (RAAS) caused by high salt intake, contributes to the underlying physiology of insulin resistance in humans [5]. The metabolites of RAAS, renin or renin-like activity, angiotensinogen, angiotensin-converting enzyme, angiotensin II (Ang II), and AT1/AT2 receptor, have been localized to adipocyte of both rodents and humans [6,7]. The release of Ang II in salt-loading adiopose tissue activates the the production of inflammatory cytokines, adipogenesis/lipogenesis, and cardiac hypertrophy, but reduces the secretion of insulin [8]. In a recent animal model, high salt-sensitivity was shown to lead to diabetes related obesity and vascular dysfunction via the activation of RAAS [9]. In a previous report, salt-inducible kinase (SIK), a novel serine/threonine protein kinase, was isolated from the adrenal glands of high-salt diet-fed rats [10]. With RAAS activation, the adipose-specific SIK2 has been shown to contribute to phosphorylation of insulin-receptor–substrate-1(IRS-1) and Akt on the early stage of insulin signaling in vitro as well as in vivo [11].

A further two possible mechanisms, a mitogen-activated protein kinase (MAPK)/extracellular signal-regulated kinase (ERK) signaling cascade and phosphinositide-3-kinase/Akt(PI3K/Akt)- dependent mammalian target of rapamycin (mTOR) signaling, were investigated in the studies previously mentioned. MAPK/ERK activated by growth factors, hormones, neurotransmitters, nutrients, and chemokines has been shown to have profound effects on the differentiation of 3T3-L1 preadipocytes or in other cellular models. This signaling involved the expression of the crucial adipogenic regulators CCAAT/enhancer binding protein-α/-β/-δ (C/EBP-α, -β, -δ), peroxisome proliferator-activated receptor gamma (PPARγ), and sterol regulatory element-binding protein (SREBP-1c) [12]. Obesity is associated with insulin resistance both in in vivo models and in humans, and the adipogenic stimuli, insulin, activates the MAPK/ERK pathway. An increased activation of Akt-mTOR, has been associated with the activation of SREBP-1c in de novo lipid synthesis [13]. In the Dahl/Salt-sensitive rat, a genetic model of hypertension with kidney disease, fed with a 4.0% salt diet for 21 days, significant upregulation of pS6/S6 and p-Akt/Akt, which are corresponding to mTORC1 and mTORC2 activities in the kidney, were shown [14]. Since mTOR signaling is a key regulator of adipose tissue biology and function including thermogenesis, and endocrine system, the disabled state of mTOR pathways might be implicated in obesity and type 2 diabetes mellitus, cancer, aging, and so on [15,16]. Adipose tissue secretes an array of hormones, adipokines, or adipocytokines that signal key organs to maintain metabolic homeostasis. Inflammatory cytokines such as tumor necrosis factor alpha (TNF-α), PAI-1, monocyte chemoattractant protein (MCP-1), interleukin-6 (IL-6), IL-1β, IL-18, leptin, and angiopoietin-like protein 2 develop adipogenesis [17]. In particular, TNF-α, IL-6, and IL-1β interfere with insulin signaling through the activation of MAPK and nuclear factor kappa-light-chain-enhancer of activated B cells (NFkB), resulting in insulin resistance [18]. M2 macrophage markers such as anti-inflammatory cytokines, adiponectin, IL-10, adipolin, and secreted frizzledrelated protein 5 are down-regulated in obesity induced disorders whilst promoting the uptake of dying adipocytes by macrophages [19]. The leptin-activated mTOR pathway triggers the secretion of cytokines and chemokines whilst increasing lipid droplets such as chemokines ligand-1/-2, tumor growth factor-β (TGF-β), and cyclooxygenase-2 (COX-2) [20]. Leptin dose- and time-dependently activated mTOR is enhanced by phosphorylation of STAT3 and Akt and downstream protein like P70S6K in intestinal epithelial cells [20]. Based on bioinformatics methods, Sun G. et al. defined the adipokine signaling pathway as linked or cross talked to mTOR signaling pathway in breast cancer [21].

The purpose of this in vitro study is to ascertain how high salt enhances obesity characteristics such as inflammatory response, insulin resistance, and what mechanism or signaling-pathways trigger adipogenesis. To elucidate the mechanism of salt-induced adipogenesis, the changes in RAAS and adipocytokines, inflammatory cytokines related insulin resistance, and signaling pathways of MAPK/ERK and Akt-mTOR were investigated in salt-loading adipocytes.

## 2. Results

### 2.1. Effect of High Salt on Viability in Pre-Adipocytes and Adipocytes

To estimate cell viability of 3T3-L1 preadipocytes, salt was treated up to 400 mM. The proliferation assay results revealed that treated salts up to 100 mM did not affect cell viability after 24 h incubation in preadipocytes and differentiated adipocytes (Figure 1A(a,b)). Since salt levels above 150 mM were toxic to adipocyte proliferation, three levels of salt treatments in this experiment were decided: 25, 50, and 100 mM. Even though DMEM media with low salt (75 mM) was used for this design, the total salt levels were 100, 125, and 175 mM in media after 25, 50, and 100 mM were added in each group. Since sodium concentration is maintained in a very narrow range of 137 to 142 mEq/L of plasma or 145 to 155 mEq/L of plasma water, a high salt level of 175 mM may cause a physiologically imbalanced condition between water and sodium.

### 2.2. Salt Induced SIK2, Na^+^K^+^-ATPase, and RAAS Signals

SIK-2, which is much higher in adipose tissue than elsewhere and is induced during adipocyte differentiation, was dose-dependently increased by salt (Figure 1B). As shown in Figure 1C, Na^+^/K^+^-ATPase was increased by the presence of high salt during the differentiation of 3T3-L1 adipocytes. However, high salt treatment (100 mM) decreased the expression of Na^+^/K^+^-ATPase. Salt generally increased the major metabolites of RAAS, ADD1, and MCR in a dose-dependent manner, although CYP11β2 and Ang II did not respond in the same way in high salt treatment (100 mM). (Figure 1D) We concluded that insulin resistance induced by high salt intake in adipocytes might be related to SIK2-induced RAAS activation.

### 2.3. Salt Stimulates Adipogenesis/Lipogesesis Genes in Adipocytes

To investigate the effect of high salt on adipogenesis in adipocytes, different salt levels, 25 to 100 mM, were treated during differentiation of adipocytes. Increased boron-dipyrromethene (BODIPY) intensity by increasing salt concentration, Triacylglyceride (TG) accumulation in adipocytes was significantly increased (Figure 2A(a,b)). Unactivated precursor form of SREBP-1c (125 kDa) was attached to nuclear envelope or endoplasmic reticulum membrances whereas the active form (86 kDa) was translocated into the nucleus binding to the site of DNA sequences. We found the nuclear active form of SREBP-1c was increased, but cytoplasmic SREBP-1c protein was significantly depressed by high salt treatment of 100 mM (Figure 2B). Increases in salt significantly increased the protein levels of PPARɣ and C/EBPα, and transcription factors of aP2, a carrier protein for fatty acids, for the adipogenesis (Figure 2C). Messenger RNA of lipogenesis genes such as ACC and FAS, measured by rt-PCR, were increased by salt (Figure 2D). In addition, excess salt decreased the phosphorylation of Thr^172^ of AMPK activation domain, suggesting a decrease in AMPK activity contributes to the reduction of lipolysis (Figure 2E).

### 2.4. Salt Enhances the Tolerance of Inflammatory Cytokines with Insulin Resistance

As shown in Figure 3A(a,b), the expression of inflammatory cytokines such as TNF-α, IL-17A, and COX-2 were significantly increased in salt-loading 3T3-L1 adipocytes, with MCP-1, and IL-6 were also released into media according to increasing salt treatment. From FACS analysis, IL-17A staining using F4/80 co-staining was increased by salt treatment of 50mM and it was decreased at 100 mM (Figure 3B(a,b)). In the protein (Figure 3C) and mRNA expression (Figure 3D) of leptin and leptin to adiponectin ratio (LAR) were increased by salt in a dose-dependent manner, otherwise, adiponectin was decreased by salt treatment. Activation of insulin-dependent IRS-1-Akt-AS160 consequence was evaluated for the insulin-resistance according to increasing salt treatment. The phosphorylation of IRS-1 and phosphorylation of Akt (Ser^473^) were relatively activated in 24 h of high salt treatment with induction of SIK2. It has been highlighted that salt-inducible kinase 2 (SIK2) is an important regulator of phosphorylation of IRS-1(Ser^794^) at the early phase of insulin signaling in adipocytes. However, both protein expression and phosphorylation activity of AS160 (Thr^642^), an Akt substrate of 160 kDa, a GTPase activating protein, were significantly decreased by increasing salts, resulting in insulin resistance (Figure 3E). Salt dose-dependently decreased phosphorylation of AMPK at Thr^172^, and the decreased AMPK activity contributed to the stimulation of insulin sensitivity.

### 2.5. Salt Activates MAPK/ERK and Akt Ser^473^-mTOR Dependent Mechanisms on the Adipogenesis

MAPK/ERK was phosphorylated by increasing salt treatment, but c-Jun N-terminal kinases (JNK) and p38 MAPK were not changed by salt-loading in adipocytes (Figure 4A). To determine whether the MAPK/ERK activation is salt sensitive, the MEK and ERK inhibitor, U0126 or PD98059, were pretreated in 3T3-L1 adipocytes for 1 or 24 h prior to induction of differentiation with or without salt. Both inhibitors repressed either MAPK/ERK activation or PPARγ expression at only 24 h prior to differentiation (Figure 4B). However, chemical inhibitors for 24 h incubation have been shown to interfere with adipocyte differentiation regardless of salt concentration such as C/EBPα and p-ERK clearance treated by U0126 as well as PD98059. It is important to know for how long, and how early chemical inhibitors were added to the cultures. We confirmed that an increase in salt-induced Na^+^/K^+^-ATPase expression increased the genes of adipogenesis triggered by MAPK/ERK pathway. Akt Ser^473^, not Akt Thr^308^, was dose-dependently activated by salt, and this pattern corresponded with the mammalian target of rapamycin (mTOR) expression (Figure 4C,D). ERK inhibitor, U-0126, also decreased Akt phosphorylation and mTOR expression. Inhibition of MAPK/ERK incubated with U0126 significantly reduced salt-induced p-ERK/ERK, PPARγ, and p-Akt/Akt ratio, but mTOR expression was not overly affected. However, the phosphorylated Akt (Ser^473^)-mTOR activation significantly increased the translocation of the active form of SREBP-1c (68 kDa) into the nucleus resulting in increased expression of lipogenesis-related genes in salt-loading adipocytes.

## 3. Discussion

In this study, we demonstrated for the first time that salt-induced SIK2 activation enhances an MAPK/ERK mediated Akt-mTOR–dependent up-regulation of adipogenetic genes expression such as PPARγ, C/EBPα, and SREBP-1c, and down-regulation of AMPK (Thr^172^). The crosstalk between MAPK/ERK and Akt-mTOR signaling in adipocytes may predict an increasing risk of obesity related to adipogenesis/lipogenesis, insulin resistance and production of inflammatory cytokines.

Studies in obese populations demonstrated that reduction in blood flow through subcutaneous abdominal adipose tissue was associated with releasing metabolites of RAAS, cardiac hypertrophy, and salt-sensitive hypertension [8,22,23]. We found that high salt intake activation of the metabolite RAAS system with SIK2 activation was similar to other studies. High salt intake (8%) for 3 weeks increased SIK2, and contributed to phosphorylation of Akt, and NAD(P)H oxidase activity in adipose tissues of Ren2 Tg female rats, which is overactive RAAS [10,24]. ERK knock-down showed an increase in the phosphorylation of Akt in the absence of Ang II, but ERK inhibition prevented the reduction of p-Akt/Akt by Ang II [25]. Integration of ERK1/2 and Akt pathways have been found in pre-adipocytes in response to diverse stimuli and can result in either positive or negative modulation of the pathway activity [26,27,28].

MAPK/ERK had profound effects on the differentiation of 3T3-L1 preadipocytes or in other cellular models, showing the expression of the crucial adipogenic regulators C/EBP-α/-β/-δ, PPARγ and SREBP-1 [29]. Obesity is associated with insulin resistance both in in vivo models and human models, and adipogenic stimuli like insulin activates the MAPK/ERK pathway [30]. The role of this pathway in normal and pathological adipogenesis has been intensively investigated. Regarding the increased insulin sensitivity displayed by ERK1^−/−^ mice, IRS-1 is phosphorylated by the ERK pathway and this serine phosphorylation exerts an inhibitory effect on the insulin signaling, contributing to a state of insulin resistance [31]. While the ERK pathway is involved in adipogenesis, displaying positive or negative effects, p38 and JNK seemed to have more restricted potential [32]. In a previous study, insulin and IGF-1 were shown to be weaker Ras-ERK activators and strong PI3K-mTORC1 activators [33]. The interaction between salt and MAPK/ERK activation may affect in differentiation and proliferation of preadiocytes, because U0126, a MAPK inhibitor, and PD98059, another MAPK inhibitor which is less efficient in blocking ERK, apparently inhibited salt-induced adipogenesis in mature adipocytes. In a previous study, U0126 or PD98059, the MAPK/ERK inhibitor, caused the phosphorylation of AMPK and also increased the cellular ADP: ATP or AMP: ATP ratios, accounting for their ability to activate AMPK [34]. ERK is regulated by Na^+^/K^+^-ATPase-dependent energy utilization and consequently AMPK activation in adipocytes [35]. Therefore, the role of ERK pathway is complex and depends on multiple parameters, while MAPK pathways are able to regulate adipogenesis.

The pathway of mTOR is a novel signaling that integrates nutrients, growth factors, energy status, and other cellular cues into a variety of anabolic processes and autophagy. In mTOR complexes, unlike mTORC2, mTORC1 is stimulated by the activation of insulin-PI3K-Akt pathways in response to growth factors, nutrients, and cellular energy status [36,37]. It has been described that mTOR is one of the downstream of the Akt/PKB pathway and Akt stimulates SREBP-1c and lipogenesis via mTOR dependent and independent pathways [13]. SREBP is an important transcription factor which activates major genes dedicated to the synthesis and uptake of fatty acids, triglycerides, sterols, and phospholipids [38]. We found that phosphorylated Akt (Ser^473^) in the carboxyl-terminal hydrophobic motif activated mTOR, not Akt(Thr^308^). Akt (Ser^473^) significantly increases the translocation of the active form of SREBP-1c (68 kDa) into the nucleus resulting in the increased expression of lipogenesis-related genes in salt-loading adipocytes. Since the agonists involved in Ras-ERK activation partially overlap with the signal to PI3K-mTORC1, insulin or insulin growth factor-1(IGF-1) are weaker Ras-ERK activators but stronger PI3K-mTORC1 activators [33]. This pattern is often influenced by positive feed forward and negative feedback loops, which function in both the Ras-ERK and PI3K-mTORC1 pathways. In addition, Akt-mTOR signaling also induces PPARγ and C/EBPβ/α, which are the master regulators of the differentiation and lipogenesis of adipocytes. However, with continuous activation of mTOR, it has a negative feedback loop from p70S6K signaling, contributing to insulin resistance [13,39,40]. Subsequent inhibition of the mTOR-p70S6K1 signaling pathway by AMPK has been shown to be indispensable for the differentiation of brown adipocytes. Therefore, the link of disabled mTOR pathway to metabolic diseases, and promising strategies of inhibiting mTOR in the prevention and treatment of obesity and its comorbidities should be considered.

Excessive intake down-regulated AMPK by phosphorylation of AMPK at Ser^485^/^491^ and activated p70S6K, which are predictive factors when considering an increased risk for onset of insulin resistance related diseases such as obesity and diabetes [41]. The most possible mechanism of AMPK-induced glucose uptake through activation of AS160 phosphorylation, was not detected in our study, but our insulin-Akt-mTOR hypothesis may explain the salt-induced adipogenesis [42]. Li et al. also reported that AMPK blocked the processing of SREBP isoforms by phosphorylating Ser^372^ to inhibit cleavage and nuclear translocation of SREBP-1c, and reducing the expression of the target gene [43]. Our results showed that the phosphorylation of AMPK Thr^172^ was significantly decreased by continuous salt-loading, and this decrease showed a significant increase of active SREBP-1c processed into nucleus. These results suggest that the potential for controlling the salt-sensitive obesity through AMPK as a target may be considered.

Salt-induced MAPK/ERK activation interacted with Akt/mTOR and consequently, with the increase of adipocytokine secretion by AMPK inhibition, resulted in insulin and leptin resistance. Adipose AMPK is an important regulator of energy homeostasis and is inhibited by insulin, diacylglycerol, and leptin [44]. The secretion of IL-17A was found to be more active in salt-loading adipocytes. Qu et al. has shown that increased expression of IL-17A also increases the expression of metabolic genes [45]. It has also been reported that paracrine agents released by high salt cause changes in the lipogenesis and lipolysis of adipose tissues by recruiting macrophages and T helper cells [46]. In a previous study, IL-17A level was higher in obese subcutaneous adipose tissue, and IL-17A differentially induced expression of inflammatory and metabolic genes, such as IL-6, TNF-α, IL-1β, and leptin [17]. Pro-inflammatory cytokines such as TNF-α, IL-6, and IL-1β interfere with insulin signaling through the activation of MAPK and NFkB, resulting in insulin- and leptin-resistance [18,20]. The leptin-enhanced mTOR pathway can trigger cytokines production with phosphorylation of STAT3 and Akt and downstream of P70S6K. Increased levels of p-Akt (activated) stimulate COX-2 that produces prostglandins in inflammatory and tumorigenic environments and mTORC1 activities [47]. Chan et al. reported that adipocyte COX-2/prostaglandin E2(PGE2)-mediated signaling involved in the development of obesity in vitro (3T3-L1 cells and human Simpson-Golabi-Behmel syndrome adipocytes), in vivo (high fat diet induced obesity rats and db/db mice) and human subjects [48]. COX-2/PGE2 pathway strongly triggers the production of pro-inflammatory cytokines such as TNF-α and MCP-1 corresponding to increasing insulin resistance [48].

In conclusion, pro-inflammatory adipocytokines by salt-loading SIK2-RAAS induction involved in MAPK/ERK-dependent Akt/PKB-mTOR activation, resulted in adipogenesis/lipogenesis, adipocyte inflammation, and insulin resistance. It is essential to notice that the integration of MAPK/ERK and Akt-mTOR signaling pathways is diverse in response to various biological conditions, such as different salt levels. Although the levels of salt treatment were not various in this study, salt 100 mM might be acted as toxic level on the expressions of Ang-II, leptin and Na^+^/K^+^-ATPase. A salt level between 50 or 100 mM may be homeostatic borderline in the balance of sodium and water. Cells might start the defense mechanism against hyperosmolality condition at 100 mM treatment due to the addition of salt levels in media (actual concentration of 175 mM). The other possibility was that the subject of method is either cell lysates or media according to the measurement property. For example, since the expression of adipogenesis was detected in cell lysates after lipid was excluded, TG levels must be analyzed in media. That’s why some of expression in adipogenic/lipogenic proteins or adipocytokines were not significantly changed at 100 mM salt. Although we did not evaluate the downstream protein of mTOR, p70S6K, salt involved in mTOR signaling induced down-regulation of phosphorylation of AMPK which is followed by activation of p70S6K. Crosstalk between the MERK/ERK and Akt/mTOR could be explained in salt-induced adipogenesis, adipocytokines secretion, and insulin resistance. As a new insight of adipogenesis, PPARr crosstalks with the endothelial nitric oxide synthases (eNOS) and that PPARr-altered eNOS activity may be responsible for hypertension and insulin resistance in salt-linked obesity [49]. In future, ascertaining whether a defense mechanism against high dose of intracellular salts provokes signaling for adipocytes differentiation or interacts with surrounding tissues through another pathway will be explored.

## 4. Materials/Subjects and Methods

### 4.1. Materials

Antibodies for Akt, phospho-Akt (Ser^473^), AMP-activated protein kinase (AMPKα), phospho-AMPKα (Thr^172^), the Akt substrate regulating GLUT4 translocation (AS160), phospho-AS160 (Thr^642^), ERK1/2, phospho-ERK1/2 (Thr^202^/Tyr^204^), p38 mitogen-activated protein kinase (p38), phospho-p38 (Thr^180^/Tyr^182^), IRS-1, IL-6, MCP-1, Na^+^/K^+^-ATPase, TNF-α, and α-Tubulin were purchased from Cell Signaling (Danvers, MA, USA). Ang II, IL-17A, COX-2, c-Jun-N-terminal kinase (JNK), phospho-JNK (Thr^183^/Tyr^185^), mineralocorticoid receptor (MCR), PPARɣ, C/EBPα, adipocyte protein (aP2), adiponectin, β-actin, α-adducin-1(ADD1), cytochrome P450 family 11-subfamily β-2 (CYP11β-2), and glyceraldehyde-3-phosphate dehydrogenase (GAPDH) were purchased from Santa Cruz (Dallas, Texas, USA). Antibodies for leptin, mTOR, and SREBP-1c were obtained from Abcam (Cambridge, MA, USA). For rt-PCR, the primers of target cDNAs such as fatty acid synthase (FAS) and acetyl-CoA carboxylase (ACC) were purchased. (GenoTech Corp, Dajeon, Korea) ELISA kits for detecting the cytokines such as MCP-1, IL-6, and TNF-α were purchased from BioLegend Inc (San Diego, CA, USA). U0126 or PD98059, f MEK/ERK inhibitors were purchased from Sigma Aldrich (St. Louis, MO, USA)

### 4.2. Experimental Design for the Cell Culture

For the single culture of 3T3-L1 pre-adipocytes, cells were maintained at 37 °C in a humidified 5% CO_2_ atmosphere in Dulbecco’s Modified Eagle’s Medium (DMEM) with Low-salt (NaCl 75 mM, LM001-09, Welgene Inc, Seoul, Korea) and 10% calf serum and 1% penicillin/streptomycin were added. After post-confluency (day-0), the cells were stimulated to differentiate by MDI on day-2. Cells were then maintained in 10% DMEM medium with 10 μg/mL insulin on day-4, culturing with 10% fetal bovine serum (FBS) and 1% of penicillin/streptomycin in DMEM until analysis. To investigate the in vitro effects of salt, a concentration of salt (25, 50, and 100 mM) were added to the culture medium every two days during differentiation. For experiments performed with MEK/ERK inhibitors, U0126, or PD98059, post-confluent 3T3-L1 preadipocytes were preincubated for one or 24 h, respectively, prior to induction of differentiation.

### 4.3. Cell Viability and Accumulation of Lipid Droplets

Cell viability was assessed using the cell counting kit (CCK-8) (Dojindo Lab, Tokyo, Japan). Briefly, pre-adipocytes were seeded onto 96 wells and cultured in the presence of various concentrations of salt (0 to 400 mM). CCK-8 solution was added and incubated at 37 °C for 4 h. The absorbance was read at 450 nm on a microplate reader (Bio-Rad Life Science, Hercules, CA, USA). Using confocal immunofluorescent analysis, lipid droplets in 3T3-L1 adipocytes were labeled with BODIPY^®^ 493/503 (green), and fluorescent DNA dye (blue). The fluorescence intensity ratio showed the concentration dependence of salt induced lipid accumulation normalized to the number of cells. The signal visualized, and digital images were obtained using a Zeiss LSM 510 confocal microscope (Carl Zeiss Micro Imaging Inc., Jena, Germany).

### 4.4. Quantitation Polymerase Chain Reaction (PCR) and Western Blotting Analysis

For gene expression analyses, total RNA extraction from the cell using TRIzol (Invitrogen, Grand Island, NY, USA) and was reverse transcribed into synthesized cDNA using superscript III first-strand synthesis system kit (Invitrogen). SYBR Green-based quantitative real-time PCR (rt-PCR) was performed by CFX Connect Real-time system (Bio-Rad^®^, CA, USA). For the protein preparation in adipocytes, after homogenizing fat tissues or lysing cells with RIPA (150 mM NaCl, 1% NP-40, 0.5% deoxycholic acid, 0.1% SDS, 50 mM Tris-Cl, pH 7.5), samples were kept on ice for 1 h, and then centrifuged at 4 °C for 15 min up to14,000 RPM. After centrifuging, neither the fat on the top nor the pellet in the bottom was collected, and the middle layer was collected for the western blot. Total protein concentration was almost 20 mg per 4 × 10^4^ cells. Proteins (20 to 35 μg) from adipocytes were separated from 8% to 15% SDS-PAGE gel, and transferred to nitrocellulose membranes and incubated with the indicated antibodies and horseradish peroxidase-coupled anti-species antibodies. Proteins were visualized by Bio-Rad chemiluminescence system (#17001401, CA, USA) and all western data shown throughout were from an average of more than three separate experiments.

### 4.5. ELISA and Intracellular Cytokines Staining

Measurement of cytokines production, TNF-a, IL-6, and MCP-1 in culture supernatants were determined using a commercially available ELISA kit (BioLegend, San Diego, CA, USA). After rabbit anti-mouse TNF-a, IL-6, and MCP-1 were incubated for 1 h, goat anti-mouse TNF-α, rat anti-mouse IL-6, and hamster anti-mouse MCP-1 IgG HRP conjugates were added and incubated. This was terminated with stop buffer, and the color change was measured at 450 nm using a microplate reader.

To assay the production of intracellular cytokines like IL-17A, the plates with Golgi plug (BD, San Jose, CA, USA) were incubated for 6 h. After incubation, cells were preincubated on ice for 5 min with anti-mouse 2.4G2 (145-2C11) (BD PharMingen, San Diego, CA, USA) to block the Fc receptors for incubation for 15 min at 4 °C. The cells were incubated with anti- monoclonal antibody of CD11b and anti-F4/80 (BD PharMingen, San Diego, CA, USA). For staining, the cells were resuspended and incubated with 250 µL of fixation/permeabilization solution (BD GolgiStop^TM^ protein transport inhibitor, San Jose, CA, USA) for 20 min at 4 °C. After washing, the staining of intracellular cytokines performed with anti-mouse IL-17A, and cells incubated for 30 min at 4 °C. Fluorescence-activated cell sorting (FACS) analysis was conducted by Sony LE-SH800 flow cytometer (Sony Biotechnology Inc, San Jose, CA, USA). FACS data were analyzed using SH800 software version 1.6.

### 4.6. Statistical Analysis

Data are presented as mean ± standard deviation (SD). Statistical difference was determined by t-test or one-way ANOVA by using the Microcal Origin Software (version 6.0) or SPSS 12.0 (SPSS Inc., Chicago, IL, USA). Statistical significance was set at *p*-value < 0.05.

## Figures and Tables

**Figure 1 ijms-20-00160-f001:**
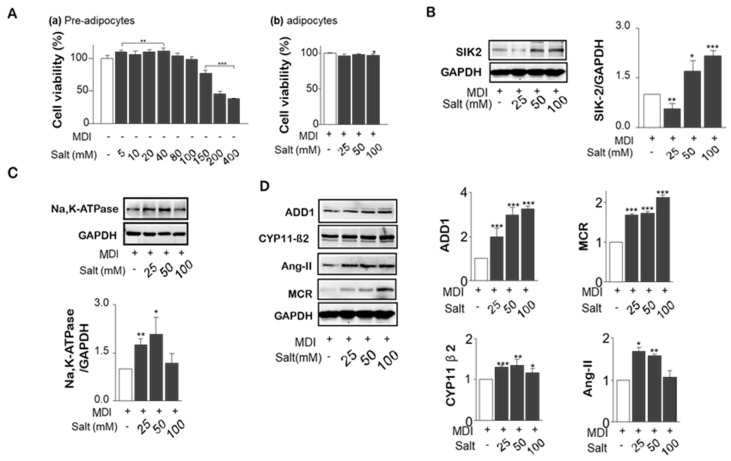
Effect of different levels of salts on cell viability, the expression of SIK2 and Na^+^/K^+^-APTase and insulin-related renin-angiotensin-aldosterone system (RAAS) signaling. 3T3-L1 preadipocytes were treated with various doses of salt (0 to 400 mM) during the differentiation period (from day 0 to day 7). Cell viability was measured by the CCK-8 kit in both of preadipocytes (**A**-**a**) and adipocytes (**A**-**b**). Protein expression of SIK2 (**B)** and Na^+^/K^+^ -ATPase (**C**) in salt-loading adipocytes were measured by western blotting. The major metabolites of RAAS, ADD1, CYP11β2, Ang II, and mineralocorticoid receptor (MCR), were analyzed by western blotting (**D**). Data are expressed as mean ± SE of three independent experiments or more. * *p* < 0.05, ** *p* < 0.01, *** *p* < 0.001, significant difference vs control.

**Figure 2 ijms-20-00160-f002:**
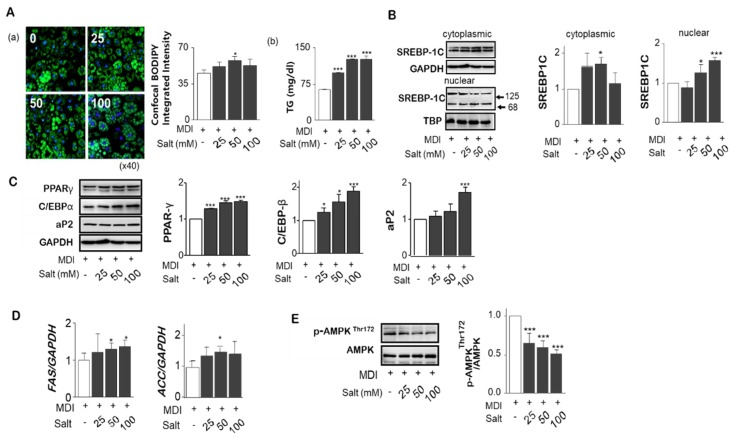
Effect of different levels of salts on TG accumulation, gene expression related to adipogenesis and lipolysis in adipocytes. Using confocal immunofluorescent analysis of 3T3-L1 adipocytes, lipid droplets have been labeled with BODIPY^®^ 493/503 (green), and fluorescent DNA dye (blue). The fluorescence intensity ratio shows the concentration dependence of salt induced lipid accumulation normalized to the number of cells. (**A-a**) Triglyceride levels in culture supernatant of salt treated 3T3-L1 adipocytes. (**A-b**) The nuclear active form of SREBP-1c was increased, but cytoplasmic SREBP-1c protein was significantly depressed by high salt treatment of 100 mM. (**B**) Protein expression for PPARɣ and C/EBPα and transcription factors of aP2 related to adipogenesis were detected by immunoblotting assay. (**C**) Messenger RNA of lipogenesis genes, fatty acid synthase (FAS) and acetyl-CoA carboxylase (ACC), were measured by rt-PCR. (**D**) Total- and phospho-AMPK^Thr172^ were analyzed by western blotting. (**E**) Data are expressed as mean ± SE of three independent experiments or more. * *p* < 0.05, ** *p* < 0.01, *** *p* < 0.001, significant difference vs control.

**Figure 3 ijms-20-00160-f003:**
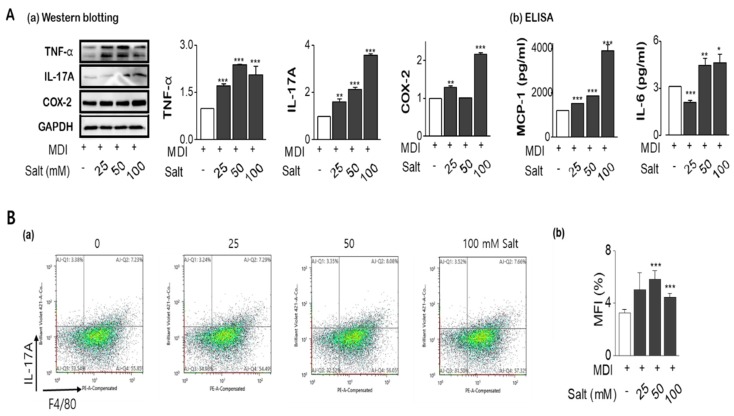
Salt modulates the production of pro-inflammatory cytokines and insulin resistance in adipocytes. Adipocytes were pretreated with concentrations of salt (0 to 100 mM) during differentiation of adipocytes. Pro-inflammatory cytokines were detected by western blotting (**A-a**), ELISA (**A-b**), and flow cytometry (**B-a**,**-b**). The leptin and adiponectin, and leptin/adiponectin ratio (LAR) were analyzed by western blotting (**C**) and semi-quantitative rt-PCR (**D**). Cell lysates were immunoblotted with antibodies against the insulin signaling such as IRS-1, phospho-IRS-1 (Ser^636/639^), AS160, and phospho-AS160 (Thr^642^). (**E**) Data are expressed as mean ±SE of three independent experiments or more. * *p* < 0.05, ** *p* < 0.01, *** *p* < 0.001, significant difference vs control.

**Figure 4 ijms-20-00160-f004:**
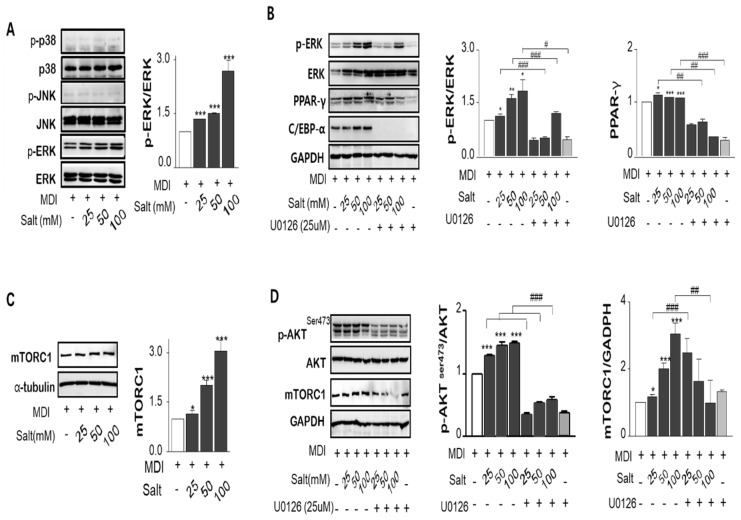
Salt regulates MAPK/ERKs and Akt-mTOR pathways. The ratio of phospho-ERK vs total ERK, phospho-p38 vs total p38 and phospho-JNK vs total JNK proteins were measured by immunoblotting. (**A**) U0126, inhibitor of MAPK and ERK, depressed the salt-induced up-regulation of p-ERK/ERK1/2, and PPARɣ. (**B**) Post-confluent 3T3-L1 preadipocytes were preincubated by MEK/ERK inhibitor for 24 h prior to induction of differentiation. The ratio of phospho-Akt^Ser473^/Atk protein levels expression were increased by increasing salt treatment, but phosphorylation at Akt^Ser308^ was not involved. The inhibitor of MAPK/ERK, U0126, declined the salt-induced Akt activation and mTOR protein expression. (**C**,**D**). Data are expressed as mean ± SE of three independent experiments or more. * *p* < 0.05, ** *p* < 0.01, *** *p* < 0.001, significant difference vs control.

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
