# Peer review of "Salt Induces Adipogenesis/Lipogenesis and Inflammatory Adipocytokines Secretion in Adipocytes"

_ijms, 2019, doi:10.3390/ijms20010160_

Reviewer 1 Report

In this paper Lee, M. et al analyze the adipocytes molecular responses to salt treatment in order to explain how high salt dietary intake can enhance metabolic syndrome. Different cytokines, proteins and pathways were considered. In my opinion this work is complete enough and clearly presented. To strength the significance of the work, I suggest to explain also that PPARgamma crosstalk with the important endothelial isoform of NItric Oxide Synthases (eNOS) and that any change in PPARgamma could be also responsible of a dangerous eNOS decreased activity (see Maccallini, C. et al The Positive Regulation of eNOS Signaling by PPAR Agonists in Cardiovascular Diseases (2017) American Journal of Cardiovascular Drugs, 17 (4), pp. 273-281).

Author Response

Reviewer 1

In this paper Lee, M. et al analyze the adipocytes molecular responses to salt treatment in order to explain how high salt dietary intake can enhance metabolic syndrome. Different cytokines, proteins and pathways were considered. In my opinion this work is complete enough and clearly presented. To strength the significance of the work, I suggest to explain also that PPARr crosstalk with the important endothelial isoform of NItric Oxide Synthases (eNOS) and that any change in PPARr could be also responsible of a dangerous eNOS decreased activity (see Maccallini, C. et al The Positive Regulation of eNOS Signaling by PPAR Agonists in Cardiovascular Diseases (2017) American Journal of Cardiovascular Drugs, 17 (4), pp. 273-281).

Ans> Even though PPARr activation is tissue-dependent according to different ligands, your suggestion may arise another possibility of explaining the mechanisms of salt-induced adipogenesis. It was discussed with reference at line 377 of page 10. (#50) Since this is in vitro study using only 3T3-L1 cells, not co-culture with endothelial cells, or not eNOS -/- transfected cells, I will consider your suggestion that salt induced adipogenesis might be caused by the crosstalk between PPARr and eNOS activity during adipocytes maturation in the future.   

 Reviewer 2 Report

This manuscript described the effect of salt-loading on adipogenesis, lipogenesis, RAAS signaling, and inflammatory cytokines expression in adipocytes. The authors reached the conclusion that salt-induced SIK2 activation enhances adipogenic gene expression by MAPK/ERK and Akt-mTOR dependent mechanisms. The manuscript is well written and presented data are convincing. I have no critical comments and recommend to accept the paper for publication in Int J Mol Sci.

Author Response

Reviewer 2

This manuscript described the effect of salt-loading on adipogenesis, lipogenesis, RAAS signaling, and inflammatory cytokines expression in adipocytes. The authors reached the conclusion that salt-induced SIK2 activation enhances adipogenic gene expression by MAPK/ERK and Akt-mTOR dependent mechanisms. The manuscript is well written and presented data are convincing. I have no critical comments and recommend to accept the paper for publication in Int J Mol Sci.

Ans> I appreciated your kind review.

Reviewer 3 Report

The authors investigated the molecular mechanism of adipogenesis in salt-related adipocytes. Salt treatment increased lipid accumulation and expression of adipogenic genes. Moreover, the expression of pro-inflammatory adipocytokines was also enhanced by salt treatment. MAPK/ERK, Akt-mTOR and adipocytokines are involved in this regulation. The data are sound. However, there are concerns that should be addressed.

 1. The methods for protein preparation and concentration should be included.

2. 100 mM salt did not affect to cell viability in adipocytes. Does high concentration of salt (100 mM) affect to the cell condition? Cell morphology was changed? The enlarged pictures of adipocytes should be shown.

3. Activation of AMPK, Akt, and AS160 are clearly changed by salt treatment. However, the changes of TG level and expression of adipogenic and lipogenic proteins were so little. This discrepancy should be clearly explained.

4. English should be revised. There are grammatical errors. Prior to submission, please check carefully.

Author Response

Reviewer 3

The authors investigated the molecular mechanism of adipogenesis in salt-related adipocytes. Salt treatment increased lipid accumulation and expression of adipogenic genes. Moreover, the expression of pro-inflammatory adipocytokines was also enhanced by salt treatment. MAPK/ERK, Akt-mTOR and adipocytokines are involved in this regulation. The data are sound. However, there are concerns that should be addressed.

1. The methods for protein preparation and concentration should be included.

Ans> I added both methods of the protein preparation and concentration in adipocytes at line 137 of 3 page. The following sentences were added in manuscript. “For the protein preparation in adipocytes, after homogenizing fat tissues or lysing cells with RIPA(150 mM NaCl, 1% NP-40, 0.5% deoxycholic acid, 0.1% SDS, 50 mM Tris-Cl, pH 7.5), samples were kept on ice for 1 hour, and then centrifuged at 4oC for 15 mins at 10,00014,000 RPM. After centrifuging, neither the fat on the top nor the pellet in the bottom was collected, and the middle layer was collected for the western blot. Total protein concentration is almost 20 mg per 4X104 cells.”

2. 100 mM salt did not affect to cell viability in adipocytes. Does high concentration of salt (100 mM) affect to the cell condition? Cell morphology was changed? The enlarged pictures of adipocytes should be shown.

Ans> First of all, the cell viabilities were measured in preadipocytes to decide salt levels of treatments. (Fig 1A) Since the levels of salt 150mM or more reduced cell viabilities, salt levels of treatments were decided up to 100mM. The cell morphology was also not changed up to salt 100mM. The Fig 2A pictures from confocal BODIPY were detected in mature adipocytes. Currently, I have only pictures of X40 and X20. I changed pictures of X40 instead X20 in the revised manuscript. However, the bar graph for fluorescence intensity ratio, TG per DNA, was correct.      

3. Activation of AMPK, Akt, and AS160 are clearly changed by salt treatment. However, the changes of TG level and expression of adipogenic and lipogenic proteins were so little. This discrepancy should be clearly explained.

Ans> The reasons that the expression of adipogenic and lipogenic proteins in some cases were slightly increased in 50 and 100mM salt loading cells are followings.

Firstly, salt level between 50 or 100mM may be homeostatic borderline in the balance of sodium and water. Cells might start the defense mechanism against hyperosmolality condition at 100mM treatment due to the addition of salt levels in media (actual concentration of 175mM). (line 373 at page 10) That’s why some of expression in adipogenic and lipogenic proteins were so little changed at 100mM salt. Moreover, our unpublished in vivo study also had not been shown dose-dependent expression for adipogenesis at the highly salty diet (8%). We assumed the results from the crosstalk between the adipocytes and the other organs such as liver, heart, kidney and blood, affected the condition of which metabolism will come earlier than salt homeostasis.

Secondly, in my study, subject of method is either cell lysates or media according to the measurement property. For examples, since the expression of adipogenesis was detected in cell lysates after lipid was excluded, TG levels must be analyzed in media. Similarly, western blotting method for adipokines could be done in cell lysates, but some of adipokines was detected by ELISA in media. That’s why all adipogenic and lipogenic proteins or mRNAs showed large increases or little increases.

4. English should be revised. There are grammatical errors. Prior to submission, please check carefully.

Ans> Thank you very much. Since my original manuscript had been reviewed by the American English Editing Co., I submitted without my second proof reading. Also, one of authors, Sungbin Richard Sorn, is a native speaker, and he tried not to make grammatical errors in this time.

Round  2

Reviewer 1 Report

Revised Manuscript was improved and, in my opinion, can be accepted for publication.

Reviewer 3 Report

My concerns were improved. I have no further comment.